# Exploring the Psychological Nexus of Virtual and Augmented Reality on Physical Activity in Older Adults: A Rapid Review

**DOI:** 10.3390/bs14010031

**Published:** 2023-12-31

**Authors:** Andrea Chirico, Marco Avellone, Tommaso Palombi, Fabio Alivernini, Guido Alessandri, Lorenzo Filosa, Jessica Pistella, Roberto Baiocco, Fabio Lucidi

**Affiliations:** 1Department of Psychology Development and Socialization Processes, Sapienza University of Rome, 00185 Rome, Italy; andrea.chirico@uniroma1.it (A.C.); marco.avellone@gmail.com (M.A.); fabio.alivernini@uniroma1.it (F.A.); jessica.pistella@uniroma1.it (J.P.); roberto.baiocco@uniroma1.it (R.B.); fabio.lucidi@uniroma1.it (F.L.); 2Department of Psychology, Sapienza University of Rome, 00185 Rome, Italy; guido.alessandri@uniroma1.it (G.A.); lorenzo.filosa@uniroma1.it (L.F.)

**Keywords:** older adults, virtual reality, physical activity

## Abstract

With the global population of older adults projected to double to 2.1 billion by 2050, it becomes crucial to promote healthy aging to alleviate the associated disease burden. In this context, technology, particularly virtual reality (VR) and augmented reality (AR), has garnered attention for its potential to augment physical activity in older adults. These immersive technologies offer interactive and enjoyable exercise experiences, making physical activity more appealing. However, the effectiveness of these interventions is not solely attributed to technology itself but is deeply intertwined with psychological processes. This rapid review examines the effectiveness of VR and AR interventions in enhancing physical exercise among healthy older adults while exploring the role of psychological variables, including mood, self-efficacy, and motivation. The results of the study show that technology-enhanced physical activity interventions hold great promise but call attention to the need for a comprehensive understanding of psychological dynamics that will pave the way for more tailored and effective interventions. Future research endeavors should aim to bridge these gaps in knowledge to optimize the impact of technology on healthy aging.

## 1. Introduction

In recent years, many advances have been made regarding medicine, technology, and socioeconomic development, and the knowledge of the general public about health-related factors has vastly improved. As a result, innovative therapies and treatments have been successfully implemented in healthcare, leading to an increase in life expectancy across the world. In fact, by the end of the decade (2030), the number of people in the world aged 60 years and older will be 34% higher, increasing from 1 billion in 2019 to 1.4 billion. By 2050, the global population of older people is estimated to double up to 2.1 billion [1]. Aging starts in the later part of the life cycle, around 60 years of age [1]. This period is associated with the emergence of several health-related states, commonly called geriatric syndromes, which are often the consequence of multiple underlying factors and include an increased risk for debilitating conditions such as frailty, delirium, dementia, and cancer, although the most frequent issues faced by individuals aged 65 and over are non-pathological age-related changes, including normal declines in cognition, physical limitations, and loss of partners and friends [2]. With it being a particularly frail condition, in which many factors (i.e., socio-economic and environmental) may contribute to the aggravation of physical and mental health, now more than ever aging represents a challenge to health institutions in which the active and healthy aging policy assumed a fundamental role.

According to the World Health Organization (2020; WHO), healthy aging consists of developing and maintaining the functional ability that enables physical and psychological well-being in older individuals. This ability is strongly influenced by the social and economic resources and opportunities available to people across their life course (e.g., social and family support, health insurance, and high income). Maintaining certain healthy behaviors throughout life contributes to many favorable outcomes regarding older adults’ health conditions [2]. In particular, physical activity stands as a cornerstone in supporting healthy aging, with its primary aim being the maintenance or enhancement of individuals’ functional abilities. This is especially crucial in addressing sedentary lifestyles while concurrently fostering an increased sense of accomplishment, self-assurance, and other social benefits. Physical activity programs yield numerous advantages, such as decelerating degenerative processes, enhancing physical capabilities, and positively influencing functional autonomy and overall quality of life [3,4]. Engaging in physical activity regularly has been linked to numerous health benefits, including reduced risk of obesity, cardiovascular diseases, cancer, and overall mortality [2]. Moreover, PA can enhance functional and cognitive capacities, improve psychological well-being, and increase the likelihood of successful aging [5].

Although the benefits of physical activity in healthy aging are widely recognized, the latest global estimates of the WHO showed that 1.4 billion adults (27.5% of the world’s adult population) fall short of reaching the prescribed standard of physical activity, which tends to decrease among both women and men as they get older [6]. Given these premises, it is important to understand the individual variables that could promote physical activity and healthy lifestyles in the older adult population.

While face-to-face interventions have been evaluated and reported in the scientific literature [7], the opportunities coming from the innovation of new technologies attracted some researchers to evaluate the use of technology to prompt physical activity in older adults [8,9]. Research on the utilization of internet-based resources and new technologies among older adults has shown more favorable attitudes when compared with a decade ago [10]. Additionally, there has been an increase in technological proficiency and openness to innovation within this demographic [11]. These findings support the feasibility of introducing technology-based interventions aimed at fostering physical activity or exercise in older adults in order to prevent health issues [12].

Among the newest digital tools, virtual reality (VR) has gained interest within different health contexts [13,14]. VR creates immersive virtual environments in which the body, environment, and brain are closely related. The simulated environment can be easily manipulated, facilitating experimental tasks that are difficult to implement in real-world settings. Other benefits of VR regard the possibility to participate in potentially dangerous tasks, such as moving in a complex environment or driving a car, in a controlled ecological setting [11], thus allowing one to perform a task safely [12,13,14,15,16]. The virtual environments produced by VR devices can be portrayed as a blend of tangible and computer-generated components, forming a spectrum known as the reality–virtuality continuum. Within this mixed reality setting, elements from both the physical world and the virtual realm are employed, giving rise to concepts like augmented reality and augmented virtuality. In the scientific literature, three distinct categories of VR systems are documented, each providing varying levels of immersion: (1) non-immersive VR systems involve the creation of a 2D virtual environment displayed on a computer screen, as seen in serious video games; (2) immersive VR systems deliver a comprehensive simulated experience through the utilization of various devices, including head-mounted displays, audio equipment, and haptic devices; and (3) semi-immersive systems deliver a mixed reality experience using a stereo image of a 3D simulated environment, which is displayed on a large monitor and adjusted based on the observer’s head position.

Augmented reality is a technological paradigm that overlays digital information or virtual elements onto the real-world environment, enhancing the user’s perception and interaction with their surroundings. Unlike the immersive, fully simulated environments of VR, AR integrates digital elements into the user’s existing reality, creating a blended experience. In the context of healthcare and physical activity promotion, AR operates on the theoretical foundation of enhancing real-world experiences through digitally augmented elements. The theoretical underpinning lies in the potential of AR to seamlessly integrate with daily life, providing users with contextual and personalized information that can influence behavior and decision-making.

These technologies offer opportunities for interactive and enjoyable exercise experiences, making physical activity more appealing and engaging for older adults [15]. VR and AR have shown potential in improving recovery from illness, physical function, balance, and mobility among older adults [16]. In addition, the enjoyable yet challenging nature of VR activities may play a significant role in providing valuable feedback to older adults, as well as enhancing motor learning and adherence to exercise programs [17]. These strategies have shown promising outcomes and contribute to the overall well-being of older individuals; however, it is not clear how the psychological variables could contribute to fostering digital interventions and if this issue has been properly addressed by the literature.

The present study aims to provide a rapid review of the literature regarding the effectiveness of interventions employing VR (either immersive or non-immersive) or AR technology to improve physical exercise in healthy older adults (>60 years), investigating psychological predictors (e.g., mood) or outcomes (e.g., well-being) of engagement in the physical activity. To comprehensively address the research objectives within a limited time frame, a rapid review approach was chosen for this study. Rapid reviews are particularly suitable when timely insights are needed, as in the case of emerging technologies and interventions. The rapid review method was chosen to efficiently gather and present relevant insights into the effectiveness of VR and AR interventions in promoting physical exercise among healthy older adults, recognizing the need for timely information in the rapidly evolving landscape of technology-based healthcare interventions. The choice of a rapid review methodology comes with inherent advantages and limitations. On the positive side, rapid reviews allow for a quicker synthesis of existing evidence, making them well suited for informing timely decision-making and policy development. The streamlined process involves more focused search strategies and inclusion criteria, facilitating a faster turnaround [18]. On the other hand, the expedited nature of the process may result in a trade-off between speed and comprehensiveness. While we aim to capture the essential findings within the available time frame, the condensed approach may not cover the breadth of literature that a traditional systematic review would [18]. The decision to opt for a rapid review, as opposed to a standard systematic review, was driven by the urgency of understanding the current state of the literature on the effectiveness of VR and AR interventions in promoting physical exercise among healthy older adults, given the rapidity of the advancement of technology.

## 2. Materials and Methods

The rapid review process was designed to be completed within a condensed time frame, balancing the need for timely results with methodological rigor. While a standard systematic review often takes several months to years, the rapid review was conducted within a more expedited schedule; specifically, it took three full months, recognizing the rapid advancement of the technology devices. This study was conducted according to PRISMA guidelines, which report the preferred reporting items for rapid reviews [19]. To identify the papers for this rapid review the authors searched the relevant literature in the Scopus and Medline databases from 2000 to 2023, considering the technological advancements in VR and AR. This limited time frame was chosen to ensure relevance to the rapidly evolving field. Search terms were as follows: “older” AND “adults” OR “elderly” AND “physical” AND “activity” OR “physical” AND “exercise” OR “exercise” OR “fitness” AND “virtual” AND “reality” OR “augmented” AND “reality”. This strategic choice aimed to streamline the search process while capturing the essence of the study’s focus. In line with Cochrane rapid review methods recommendations [18], before performing the screening of the studies, the authors used standardized titles and abstract forms with the same pool of studies for the entire research group to calibrate and test the review form. The first and the second authors independently screened 80% of titles and abstracts, with conflict resolution. Then, the first author screened the remaining abstracts, and the second author screened all excluded abstracts and if needed resolved conflicts. Relating to the full-text inclusion, a standardized full-text form was used in order to test and calibrate the review form. Then, the first author screened all included full-text articles, and the second author screened all excluded full-text articles. The present screening procedure was performed using the Rayyan.ai tool [20]. All tables and figures were uploaded to OSF at the following link: https://osf.io/fqas4/.

Table 1 shows the summary of the inclusion and exclusion criteria of the study characteristics based on the PICOS framework (i.e., populations/participants, interventions and comparators, outcome(s) of interest, and study designs/type) [18,21]. Peer-reviewed articles written in the English language were included. Key stakeholders were involved and consulted during the review process in order to set and refine the review question, eligibility criteria, and the outcomes of interest.

### 2.1. Data Extraction

The first and second authors checked for the correctness and completeness of the extracted data. For each selected study, socio-demographic (e.g., sample mean age), methodological (e.g., VR/AR software employed, hardware, a primary and secondary outcome, PA procedure for experimental group [EG] and control group [CG], and assessment of PA and psychological variables), and statistical variables (e.g., data analysis and significant results) were extracted.

#### Risk of Bias Assessment

The risk of biases was assessed through a checklist derived from integrating the quality assessment tool for quantitative studies and the Cochrane Collaboration’s tool for assessing the risk of bias [22,23].

The tool used in the present rapid review assessed the following potential areas of bias: (1) randomization process, (2) deviations from the intended interventions, (3) missing outcome data, (4) measurement of the outcome, and (5) selection of the reported results.

The second author rated the risk of bias in the included studies, with the first author fully verifying all judgments (and support statements). Interrater agreement was excellent (r = 0.9). The final score identified whether a study was either at low, moderate, or high risk of bias.

## 3. Results

### 3.1. Database Searching

Database searching yielded a total of 346 abstracts. Of these, n = 2 duplicates were excluded. Then, n = 342 records were independently screened for the abstract, and n = 271 were excluded because of the defined exclusion criteria: wrong study population (40%), wrong study design (21%), wrong outcome (36%), or wrong publication type (3%). Therefore, the n = 71 remaining records were screened for the full text, and n = 36 were excluded for the following reasons: did not meet the inclusion criteria for age population, wrong study design, wrong outcome, did not employ VR/AR technology, or foreign language. Finally, n = 35 records were included in the review (see Figure 1 for the flow diagram).

Table 2 provides a synoptic summary of the search results.

### 3.2. Risk of Bias Assessment

Most of the studies did not follow strictly methodological criteria, showing a moderate overall risk of bias. Considering the Cochrane Collaboration’s tool for assessing the risk of bias [22,23], only the study of Park and Yim [31] was evaluated as having an overall “low risk” of bias fulfilling all the defined criteria. Considering all the criteria screened following the procedure (i.e., 1: randomization process; 2: deviations from the intended interventions; 3: missing outcome data; 4: measurement of the outcome; 5: selection of the reported results), the studies were evaluated as follows:(a)27 studies (77%) were evaluated as having moderate risk of bias; 7 studies (20%) were evaluated at high risk.(b)Considering each of the domains evaluated:a.Randomization process: Seven studies (20%) were evaluated as “low concerns”, 22 studies (62.7%) were evaluated as “some concerns”, and 6 (17.3%) were evaluated as being at “high risk” of bias.b.Deviation from intended intervention: Ten studies (8.5%) were evaluated as “low concerns”; twenty-four studies (68.4%) were evaluated as “moderate risk”, and one study (2.85%) was evaluated as being at “high risk”.c.Missing outcome data: Four studies (11.4%) were evaluated as “some concerns”, while all the others were evaluated as being at “low risk”.d.Measurement of the outcome: Four studies (11.4%) were evaluated as “some concerns”, while all the others were evaluated as being at “low risk”.e.Selection of the reported result: Twenty-nine studies (82.65%) were evaluated as “some concerns”; six (17.1%) were evaluated as “low risk”.

The results show as more frequent a low risk of bias in terms of “measurement of the outcome” and “missing outcome data”, while most biases rely on the randomization process poorly implemented by the studies. See Figure 2 and Figure 3 for the complete evaluation.

### 3.3. Participants

The included studies evaluated a total of 1712 participants, of which 950 (55.5%) belonged to experimental groups (PA with VR/AR) and 753 (44%) belonged to the control group (traditional PA or no intervention). A total of 1063 participants were female (62%) and 633 (37%) were male. The mean age of participants was 75.12 years.

### 3.4. Study Design

Of the included articles, 68% (n = 24) consisted of randomized controlled trials (RCTs) [24,26,27,28,30,31,32,33,34,35,36,37,38,40,41,42,43,46,50,51,52,54,55]. Other studies included (N = 4) cross-over designs (alone versus with peers [48], immersive versus non-immersive [58], young avatar versus old avatar [53], and young versus old [47]), 9% were observational studies [49,56,57], two were feasibility trials [40,45], and one was based on retrospective data [25].

### 3.5. Immersive, Virtual, or Augmented Reality

Among the included studies, seven (20%) employed immersive VR [31,39,40,52,53,57,58]. Twenty-four studies (68.5%) employed non-immersive VR [25,26,27,28,31,32,33,34,35,36,37,38,42,47,48,49,50,53,54,55,56,58], while four (11.5%) employed AR technology [24,41,43,51].

Of the thirty-five studies selected, 46% (N = 16) assessed physical and psychological variables [27,32,34,35,36,37,40,41,48,50,51,52,53,54,55,56], while the others did not consider psychological variables.

### 3.6. Psychological Variables Included in the Study

Mood-related symptoms have been evaluated by four studies [37,50,53,56], a measure of the quality of life has been evaluated by four studies [36,40,54,57], two studies included a measure of satisfaction with life [52,55], two studies evaluated motivation [48,58], self-efficacy in avoid falling was evaluated by five studies [32,33,34,35,41], while two evaluated self-efficacy related to physical activity [41,53]; see Table 1 for the complete list of constructs evaluated by each study.

### 3.7. Psychological Tools Implemented

In the selected studies, mood was measured by the Psychological Health Questionnaire [59], the Geriatric Depression Scale (GDS) [60], the Beck Depression Inventory (BDI) [61], the Affect Grid [62], and the Positive Affect Negative Affect Schedule (PANAS) [63]. Quality of life, was assessed with the Short Form Health Survey 12 [64], the Short Form Health Survey (SF-36) [65], and EQ-5D-3L [66]. Motivation was assessed through the Intrinsic Motivation Inventory (IMI) [67], the Physical Exercise Adherence Questionnaire [68], and the Player Experience of Needs Satisfaction (PENS) [69]. Self-efficacy was measured trough the Falling Efficacy Scale-International (FES-I) [70], exercise self-efficacy (ESE) [71], and Tinetti falls efficacy scale [72]. For the complete list of questionnaires used in the different studies, see Table 2.

### 3.8. Neuropsychological Assessment

Cognitive status was assessed by 17% (N = 6) of the records [31,38,45,50,55,58]. Questionnaires mainly adopted were the Montreal Cognitive Assessment (MoCA) [73], Mini Mental State Examination (MMSE), Mini-Cog [74], and the MEC Spatial Presence Questionnaire (MEC-SPQ) [75].

### 3.9. Subjective Variables Related to the Device

The acceptability and usability of the employed VR and/or AR technology were measured by six of the selected articles [39,40,45,48,49,58], mainly using the System Usability Scale (SUS) [76], the Simulator Sickness Questionnaire (SSQ) [77], and the Game Experience Questionnaire [78]

Narrative results related to the outcomes of the records are presented below separately for psychological outcomes, cognitive functions, and physical and functional abilities.

### 3.10. Psychological Outcomes

#### 3.10.1. Self-Efficacy

In five out of the seven studies assessing self-efficacy, a notable improvement was observed in the intervention group(s) in comparison with the control group. Among these, four studies utilized an active non-virtual reality (non-VR) control group [26,32,35,36], and one study had a control group with no exercise ([51]. Conversely, one study did not identify significant differences between the intervention and control groups [34]. Another study had a distinct focus on investigating the effectiveness of a specific virtual reality (VR) environment, comparing outcomes between a young and old avatar [53]. All the studies implemented interventions using VR.

#### 3.10.2. Mood

Two of the five studies evaluating mood did not search the evidence for improvement in mood [37,50] but assessed depression as exclusion criteria; considering the remaining two studies, one of them found an improvement in mood after the VR intervention [56]. All the studies implemented interventions using VR.

#### 3.10.3. Quality of Life

Furthermore, six studies included measures of quality of life; two of them found significant differences between the VR intervention group and the control group [36,52]. Studies conducted by Campo-Prieto and colleagues and Garcia-Bravo and colleagues found an increased QoL in both the intervention group and the control group [40,57], while Lee found mixed results [30], and Nonino evaluated only satisfaction with life with a non-validated questionnaire [55]. All the studies implemented interventions using VR.

### 3.11. Neuropsychological Outcomes

Neuropsychological assessments were performed by four studies [31,38,50,58] of the included records. However, one evaluated global cognitive evaluation as part of the exclusion criteria [50], and the remaining found significant positive differences in favor of the intervention group compared with the control group [31].

### 3.12. Physical Activity, Balance, and Gait

Out of the 35 studies included in the review, all found significant improvement in indexes of physical activity, functionality, or balance after the intervention with immersive, non-immersive, or augmented reality but not two that were feasibility studies [49,58].

Thirteen studies included exclusively PA or balance-related outcomes, with no attention given to psychological variables [24,25,26,27,28,30,31,39,42,43,44,45,47]. Among these studies, five found significant differences regarding different measures of PA practiced with and without immersive VR by older adults. Among studies that employed non-immersive VR (e.g., motion sensors, Kinect, and the Wii Balance Board), three found significative results between groups for shoulder (flexion, abduction, and rotation joint movements) and knee (retraction, extension, and protraction) exertion using Kinect [49]; strength of bilateral knee, isokinetic peak torque, flexibility, endurance, and total work using Wii Fit Plus [26]; and muscle strength and reaction times using the Wii Balance Board [46].

### 3.13. Feasibility Studies

Three studies evaluated the feasibility of virtual reality interventions to enhance physical activity. The first study by Munoz and colleagues evaluated mainly the usability of the devices, in an observational study design [49] with a VR non-immersive system (RGBD Microsoft Kinect version 2), and assessed its effectiveness in correctly performing various exercises while also assessing their physical achievements. The degree of acceptance of the procedure was measured through a survey based on the System Usability Scale (SUS), whereas the physical performance was monitored by the system. The results showed positive outcomes in terms of usability.

The second study was a cross-over study design (immersive vs. non-immersive) by Kruse and colleagues [58] aimed to compare a traditional, recorded 2D exercise video with a VR exergame. The comparison regards enjoyment, attention allocation, perceived workload, and preference. The outcome measurements included physical functioning with different tests, the Simulator Sickness Questionnaire (SSQ) for motion sickness, the Intrinsic Motivation Inventory (IMI), and the MEC Spatial Presence Questionnaire (MEC-SPQ). The results showed slight or no differences between the two modalities (immersive vs. non-immersive).

The third study, an RCT proposed by Campo-Prieto [40], aimed to analyze the effects of an immersive VR exergaming program on physical function, quality of life, and parameters related to VR exposure. The outcome measurements included for assessing balance and gait were lower limb function, hand grip strength, the time up and go (TUG) test, and the five times sit-to-stand test (FTSTS). The results showed improvement of physical functioning and balance but no significant differences in the quality of life between groups.

## 4. Discussion

Promoting physical activity in older adults is crucial for maintaining good health and overall well-being. The scientific literature is advancing the concept of digital intervention using more often recent technologies such as Virtual Reality or Augmented Reality. As stated, there are recent reviews of the literature that provide evidence for the efficacy of the use of VR in older adults in enhancing physical activity [79].

Results on physical activity have been confirmed by the present study. Our rapid review delves into the efficacy of virtual reality (VR) and augmented reality (AR) interventions for promoting physical activity among healthy older adults, by focusing on the role of psychological factors as either mediators or moderators of intervention efficacy. Our results identified key insights, challenges, and gaps in the existing literature. Overall, most of the identified studies presented some difficulties: (i) they considered people aged 50–55 as older adults, while common research criteria establish older age as a process that begins from age 60 to 65 [80,81]; (ii) they considered PA as an outcome or a status, i.e., the difficulty encountered by an individual in executing a task or action due to old age limitations; (iii) they did not take psychological variables (e.g., mood or self-efficacy) into account for enhancing PA in older adults but rather as an outcome; and (iv) most of the studies lacked strict methodology, and the risk of a biased conclusion was high.

Our rapid review adds a timely perspective to the evolving landscape of research on technology-driven interventions for older adults. Specifically, it contributes by emphasizing the importance of considering psychological variables in the adoption and sustainability of physical activity habits through VR and AR interventions. This nuanced focus distinguishes our review from previous works and sheds light on the intricacies of technology’s impact on older adults’ physical activity.

Given this premise, in the realm of enhancing physical activity in older adults, technology stands as a promising ally. However, it is essential to recognize that its efficacy transcends the mere presence of gadgets or applications. Instead, the profound influence technology exerts on physical activity in this population is intricately tied to the underlying psychological processes it activates. However, despite the evidence of the relevance of psychological variables involved in the adoption of physical activity habits [82,83,84], it is noteworthy that the existing literature often falls short in investigating these multifaceted psychological processes. Our results showed that fewer than half of the included studies in the rapid review considered these variables; furthermore, when these variables were considered, they were often treated as secondary outcomes, focusing solely on the effects of physical activity on mental health rather than examining how they can directly influence, moderate, or mediate adherence to physical activity [85]. In other cases, these variables were only evaluated as exclusion criteria and were not considered primary outcome measures [43].

Factors such as self-efficacy, motivation, and mood emerged as central determinants in this context. As older adults engage with technology, they often experience a notable surge in self-efficacy, fostering a newfound confidence in their ability to adopt and sustain an active lifestyle. This increase in self-assurance is attributed to the opportunity to visualize their progress thanks to the technology and also to gain confidence in a safe environment [86].

Moreover, technology empowers older adults with enhanced motivation. The gamification elements incorporated into many fitness applications transform physical activity into an engaging challenge. Goal-setting features provide a clear sense of purpose, while real-time feedback and rewards offer immediate gratification, reinforcing the motivation to stay active. Additionally, the social connectivity facilitated by technology contributes significantly to mood enhancement.

A comprehensive understanding of how self-efficacy, motivation, and mood mediate the relationship between technological interventions and the intention to engage in physical activity among older adults remains an underexplored frontier. Thus, future research endeavors hold the potential to unearth the intricate dynamics at play, shedding light on the nuanced ways in which technology drives and sustains physical activity in this population.

Another of the main challenges identified throughout this literature review is the lack of standardized and defined assessment measures for psychological variables in the context of physical activity validated for the older adult population. Currently, there are no tools considered “gold standards” for the assessment of these variables in older adults, making it difficult to compare and generalize study results. This lack of consolidated assessment measures represents a significant limitation in scientific research on the subject.

Furthermore, most studies included these side effects and other health conditions in their exclusion criteria. A low incidence of adverse effects, with no impact on adherence and no significant difference between groups, was found. Our analysis suggests that AR and VR protocols, regardless of their efficacy in enhancing physical activity, balance, or other psychological variables, were generally well tolerated by older adults.

## 5. Strengths and Limitations of the Study

The rapid review methodology used in this study had advantages and trade-offs, resulting in a nuanced evaluation of the research process. One of the significant gains of opting for a rapid review was the timely acquisition of relevant insights. In a field where technology evolves rapidly, synthesizing the literature quickly helped us capture the most recent advancements and evidence on the effectiveness of virtual reality (VR) and augmented reality (AR) interventions in promoting physical activity for older adults. In addition, the rapid review allowed for a focused investigation, honing in on the literature dealing specifically with VR-based interventions for enhancing physical activity among healthy older individuals. This specificity facilitated a targeted exploration of the role of psychological factors as mediators or moderators of intervention efficacy. The discussion effectively highlighted critical challenges and gaps in the existing literature, including the limited consideration of psychological variables, methodological shortcomings in the identified studies also evidenced in some of the bias assessed, impacting mostly the sampling procedure (i.e., randomization process), and the absence of standardized assessment measures. This awareness sets the stage for future research endeavors to address these limitations. While the rapid review provides a timely overview, the condensed nature of the methodology may have limited the depth of analysis into certain aspects. Complex relationships between technological interventions and psychological variables may benefit from a more comprehensive exploration than a standard systematic review might afford. The discussion reveals that fewer than half of the included studies in the rapid review considered psychological variables. Furthermore, when these variables were considered, they were often treated as secondary outcomes, potentially overlooking their direct impact on adherence to physical activity. This limitation points to a missed opportunity for a more nuanced understanding. In summary, the rapid review approach facilitated a quick and targeted exploration of the literature, providing timely insights into the efficacy of VR-based interventions for promoting physical activity in older adults. However, the condensed methodology raised considerations about the depth of analysis and the adequacy of exploring psychological variables, in addition to categorization. These trade-offs underscore the need for a balanced approach that aligns with the specific goals and constraints of the research.

## 6. Conclusions

The present study used a rapid review approach to provide a roadmap for future research endeavors, by evaluating the role of psychological variables in the improvement of physical activity in older adults using VR or AR. Overall, our findings highlight the need for a more comprehensive exploration of psychological processes, urging researchers to delve deeper into the dynamics of self-efficacy, motivation, and mood as central determinants and not only as an outcome. Additionally, our identification of challenges, such as the lack of standardized assessment measures, offers a practical foundation for refining research methodologies (see BREQ3). Considering the actual scientific literature, our rapid review claims for further investigating the role of psychological and cognitive variables, as well as the validation of outcome measures for psychological aspects, as crucial for advancing our understanding of promoting physical activity in older adults and enhancing their overall well-being through new technologies (i.e., VR or AR). In the realm of practice, our findings underscore the potential of technology to enhance physical activity, emphasizing the need for interventions that leverage psychological factors to maximize effectiveness. Finally, our research holds potential benefits for the target population of healthy older adults. By acknowledging the influence of technology on psychological variables, particularly self-efficacy, motivation, and mood, our findings suggest that technology-driven interventions can empower older adults with increased confidence, enhanced motivation, and improved mood. These factors, in turn, may contribute to the adoption and maintenance of an active lifestyle, promoting overall well-being in the aging population.

## Figures and Tables

**Figure 1 behavsci-14-00031-f001:**
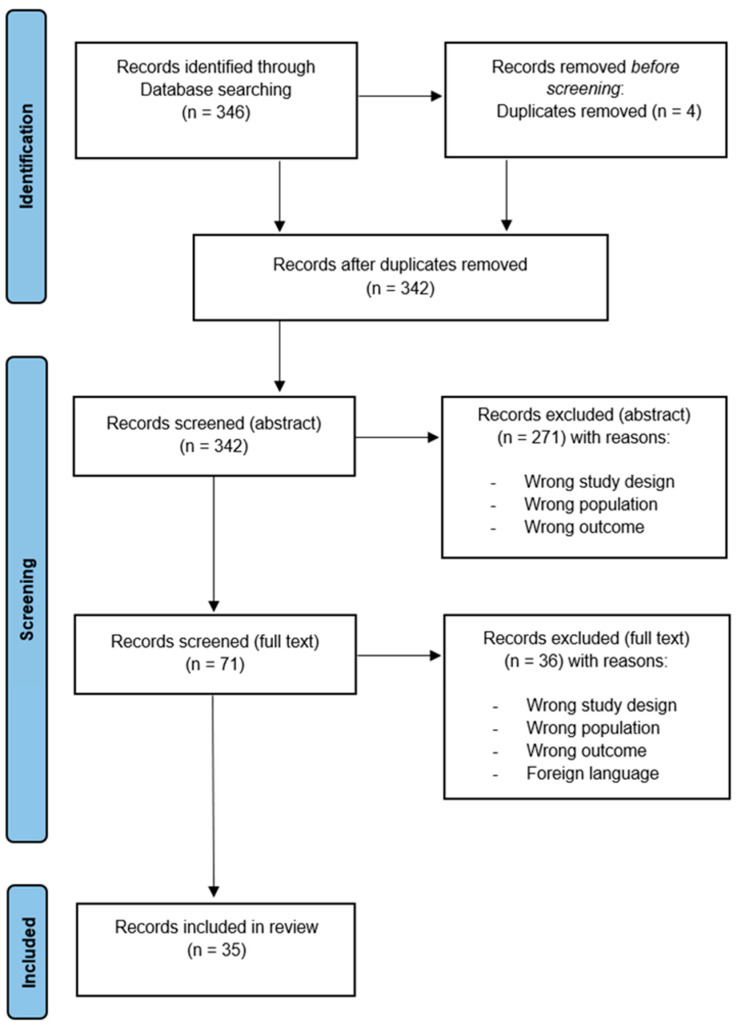
Flow diagram of the selection process.

**Figure 2 behavsci-14-00031-f002:**
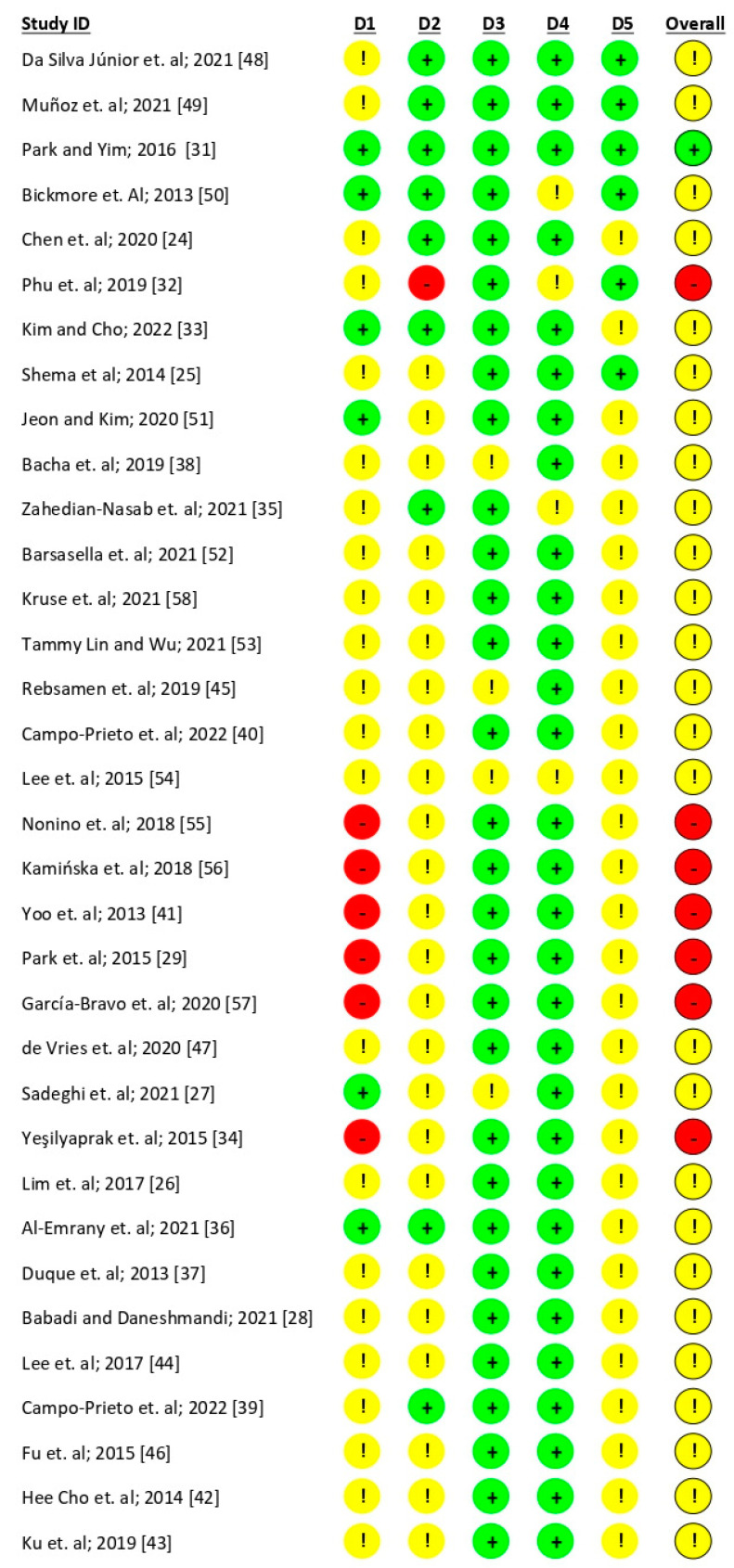
Risk of bias assessment [24,25,26,27,28,29,31,32,33,34,35,36,37,38,39,40,41,42,43,44,45,46,47,48,49,50,51,52,53,54,55,56,57,58].

**Figure 3 behavsci-14-00031-f003:**
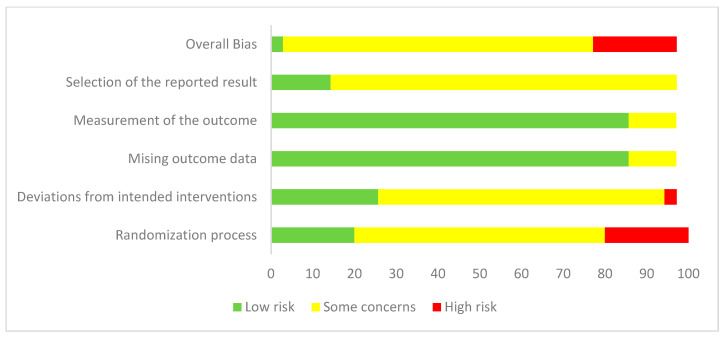
Risk of bias assessment and distribution of the studies.

**Table 1 behavsci-14-00031-t001:** Summary of inclusion and exclusion criteria according to the PICOS framework.

		Inclusion Criteria	Exclusion Criteria
P	Populations/participants	Older adults (aged 60 or over)	Adults, young adults, adolescents, or children
I	Interventions	Employing immersive or non-immersive VR or AR	None
C	Comparators	With or without a comparison group	Disease-specific protocols (e.g., Parkinson’s disease, Alzheimer’s disease, stroke), rehabilitation-specific protocols, or dual-task (cognitive and motor tasks) procedures
O	Outcome	Main outcome: elicitation or assessment of PA-related variables. Secondary outcome: psychological variables including mood, self-efficacy, and motivation.	None
S	Study designs/types	Randomized controlled trials, quasi-experimental studies, feasibility trials, cross-over, observational, prospective, systematic review	Qualitative studies, protocols, editorials, and dissertations

Note. VR, virtual reality; AR, augmented reality; PA, physical activity.

**Table 2 behavsci-14-00031-t002:** Table summarizing characteristics of the included studies.

STUDY	Ref.	DESIGN	N IG	N CG	CATEGORY	ACTIVE CG (A); NO INTERVENTION CG (B); NO CONTROL GROUP (C);	VR IMMERSIVE/VR NON-IMM/AR	DEVICE/PLATFORM EMPLOYED	VR INTERVENTION DESCRIPTION	PHYSICAL ASSESSMENT	PSYCHOLOGICAL ASSESSMENT	RESULTS AND SUMMARY
Chen et al., 2020	[24]	Prospective randomized trial: traditional tai chi exercise vs. selected AR-assisted tai chi exercises	14	14	MBA-np	A	AR	Kinect version 2 motion sensor (Microsoft Corporation, Washington, DC, USA)	Selected tai chi (sTC) exercises by using an AR training system	Functional balance: time up and go (TUG); balance: Berg balance scale (BBS); functional capacity: functional reach test (FRT); strength: lower extremity muscle strength test	na	In the IG group, BBS, TUG, and FRT scores showed significant improvement overall. Outcomes improved in both conditions, but none of them was significantly better than the other.
Shema et al., 2014	[25]	Retrospective data analysis	60	na	MBA-np	C	Non-immersive VR	VR Gait training system	VR gait training program: walking on a treadmill while negotiating virtual obstacles; three times per week for 5 weeks	Balance and functional mobility: timed “up and go” test (TUG), the two-minute walk test (2MWT), and the four square step test (FSST)	na	For the VR group, average time to complete the TUG decreased, indicating a significant reduction in fall risk. Significant improvement was observed in endurance (2MWT) and in decreased time to complete the FSST, suggesting improved functional mobility. In contrast, gait speed did not reach a statistically or clinically significant change after training.
Lim et al., 2017	[26]	Randomized comparative trial: two active VR groups: complex exercise with virtual reality (CEVR) group vs. a balance exercise with virtual reality (BEVR) group	10	10	MBA-np	A	Non-immersive VR	Wii Balance Board (Nintendo, Tokyo, Japan)	CEVR program consisted of balance, strengthening, stretching, and endurance exercises; the BEVR program consisted of the six balance exercises included in the CEVR group. Both groups were training following the Wii instructions, for 5 weeks.	Strength of bilateral knee (Biodex Multi-Joint System 4 (Biodex Medical Systems Inc., Hauppauge, NY, USA)); isokinetic peak torque and total work; balance: timed up and go test (TUG)	na	Knee extension peak torque was significantly enhanced only in the CEVR group (*p* < 0.05), but there was no difference between groups. Both groups showed significant improvement of dynamic balance measured by TUG after training sessions, but the CEVR group exhibited significant greater improvement than the BEVR group.
Sadeghi et al., 2021	[27]	Randomized controlled trial: conventional balance training (BT), VR balance training (VR), MIX: VR + BT, and pure control group	15	14 active control; 15 pure control	MBA-np	A+B	Non-immersive VR	Kinect v2 motion sensor (Microsoft Corporation)	Based on a pilot intervention, three games that challenged lower body balance: (1) the Light Race (Stomp It) mini-game from the Your Shape fitness package, (2) the Target Kick, and (3) Goalkeeper mini-games from the Sport Xbox Kinect game package	Strength: isokinetic muscle strength; balance: single-leg stance on firm and foam surfaces, tandem stance, timed up and go (TUG), 10 m walk test (10 mWT)	na	The main findings of this study were that (i) the MIX group showed the greatest improvements in strength, balance, and functional mobility compared with the other groups; (ii) the VR group exhibited better balance and functional mobility compared with the BT and the pure control groups; and (iii) the BT group exhibited better balance and functional mobility compared with the control group.
Babadi and Daneshmandi, 2021	[28]	Randomized controlled trial: VR intervention vs. conventional training vs. pure control	12	12 + 12	MBA-np	A	Non-immersive VR	Kinect v2 motion sensor (Microsoft Corporation)	The virtual reality training program included boxing, table tennis, and soccer from the sports Pack 1, as well as golf, skis, and American football from Sports Pack 2; 60 min session, three times per week, for 9 weeks	Balance: Fullerton advance balance (FAB); timed up and go (TUG), Fullerton advance balance scale (FRT); single-leg stance (SLS)	na	Improvement of balance was observed in both VR and conventional groups in right and left SLS, FRT, TUG, and FAB, but this improvement was not detected in the control group.
Park et al., 2015	[29]	Randomized controlled trial: VR intervention group vs. conventional exercise group	12	12	MBA-np	A	Non-immersive VR	Wii Balance Board (Nintendo)	Wii Fit balance exercise game/program: “Soccer Heading”, “Snowboard Slalom”, and “Table Tilt”; three times a week for 8 weeks	Balance and gait: timed up and go (TUG), sway length, average sway speed	na	In this study, both the groups showed a significantly reduced sway length and average sway speed, although the virtual reality game group showed a greater decrease. The virtual reality game group also showed an increase in TUG time.
Lee, 2021	[30]	Randomized controlled trial: VR intervention group vs. conventional exercise control group	28	28	MBA-np	A	Non-immersive VR	Virtual Active, BitGym	The VR group received non-immersive VR intervention with non-motorized treadmill (with overlooking videos) for 50 min a day for 4 weeks and 5 days a week.	Balance: one-leg-standing test (OLS), Berg balance scale (BBS), functional reach test (FRT), and timed up and go test (TUG); gait: velocity, step width, stride length, step length, with gait analyzer system	na	The changes in OLS and TUG showed a significant improvement after intervention in the VR group but not in the control group. The changes in static stride length and step length variables showed significant improvement after intervention in both groups. Both groups improved significantly in the gait components.
Park and Yim, 2016	[31]	Randomized controlled trial	36	36	MBA-p	A	Immersive VR	3-D beam projector, 3-D images of moving kayaks that were directly filmed in a river and a lake were displayed.	30 min of conventional exercise similar to the control group, and then 20 min of the virtual reality kayak program simulated with a stool and a footrest on a springboard, 20 min, two times a week for 6 weeks	Muscle strength: hand grip strength, arm curl test (ACT); balance: sitting and standing balance test	Cognitive functions: Montreal Cognitive Assessment (MoCA)	The intervention group improved significantly compared with the control group on all the measured variables (i.e., muscle strength, balance, and cognitive function).
Phu et al., 2019	[32]	Interventional study: balance rehabilitation unit (BRU) versus conventional exercise (EX) vs. pure control group	63	82 (active control); 50 pure control	MBA-p	A+B	Non-immersive VR	Wii Balance Board (Nintendo)	Balance rehabilitation unit (BRU) twice a week for 6 weeks, standing on the BRU plat- form and undertaking several exercises	Physical performance: five times sit to stand (5ST), time up and go (TUG), four square sit test (SST), gait speed; strength: hand grip strength; static tasks (i.e., eyes open firm surface (EOEA)) for 1 min	Falling efficacy: Falls Efficacy Scale-International (FES-I)	Post-intervention, both intervention groups (VR) and conventional group exercises achieved similar improvements and reported similar adherence rates and improved significantly on all outcome variables compared with the pure control group. Only the VR group improved control of static posture in the eyes closed and foam eyes closed tasks more than the conventional exercise.
Kim and Cho, 2022	[33]	Randomized controlled trial: virtual reality (VR) program vs. motor imagery training (MIT) vs. pure control	12	10 active control; 12 pure control	MBA-p	A+B	Non-immersive VR	Wii Balance Board (Nintendo)	Virtual reality exercise program, based on the following Wii available exergame: “Balance ski”, “Table tile”, “Jogging”, and “Rhythm step”, 30 min/day, 3 days/week, 6 weeks	Balance: gait view AFA-50 (alFOOTs, Seoul, Republic ofKorea)	Falling efficacy: Tinetti falls efficacy scale	The VR group showed significant higher scores at post-tests compared with the MIT group considering falling efficacy and one index of balance. Moreover, the VR and the MIT groups improved in both balance and falling efficacy significantly more compared with the pure control group.
Yeşilyaprak et al., 2015	[34]	Randomized controlled trial: VR group vs. conventional exercise group	10	11	MBA-p	A	Non-immersive VR	BTS NIRVANA VR Interactive System	Subjects were taught to follow the onscreen visual displays and to listen to audio feedback while maintaining their stability during balance activities in a standing posture.	Balance and functional capacity: Berg balance score (BBS), timed up and go (TUG), one-leg stance (OLS) test, tandem stance (TS)	Falling efficacy: Falls Efficacy Scale International (FES-I)	Berg balance score (BBS), timed up and go duration, and left leg stance and tandem stance duration with eyes closed significantly improved with time, but changes were similar in both groups. There were no significant improvements, nor significant differences between the VR group and the CG, in falling efficacy or balance in eyes open conditions following either intervention.
Zahedian-Nasab et al., 2021	[35]	Randomized controlled trial: VR group vs. conventional exercise group	30	30	MBA-p	A	Non-immersive VR	Kinect v2 motion sensor (Microsoft Corporation)	Simulating balance exercise in the game environment; selected games: Kinect Sports 1 and 2, including “penalty”, “goalkeeping”, “ski”, and “darts”; weekly basis for 6 weeks	Balance: balance and Berg balance scale (BBS), the timed up and go (TUG)	Fear of falling (FES-I)	The indices of balance among older adults improved significantly in the VR group after the intervention (BBS and TUG test). Moreover, the score of falling efficacy diminished significantly in the VR group compared with the control group.
Al-Emrany et al., 2021	[36]	Randomized controlled trial: VR group vs. conventional exercise groups	25	10	MBA-p	A	Non-immersive VR	Kinect v2 motion sensor (Microsoft Corporation)	Kinect Sports video games with the chosen option of competitive single-player mode, including “bowling”, “boxing”, “track and field”, “table tennis”, and “beach volleyball”; 60 min; 5 days weekly, 6 weeks	Balance: overall stability index (OSI) of Biodex and the functional reach test (FRT)	Quality of life (SF-36)	Results revealed significant improvement of OSI and FRT after VRT. Also, the results demonstrated significant improvements in levels of general health quality of life after the VR compared with the control group.
Duque et al., 2013	[37]	Randomized controlled trial: VR balance intervention with the balance rehabilitation unit (BRU) versus conventional exercise control group	30	40	MBA-p	A	Non-immersive VR	Balance rehabilitation unit [BRU]	Visual–vestibular rehabilitation and three different postural training VR games (maze, breakfast, and surfing) with increasing levels of complexity (maximum of 15 levels); two sessions per week, 6 weeks	Parameters measured thanks to the BRU: limits of stability (LOS), interpreted as the maximum sway on the platform before losing stability, and area of the ellipse of confidence (CE) for the center of pressure (COP); falls: number of falls in the last period	Depression: Geriatric Depression Scale (GDS); falling efficacy: Tinetti falls efficacy scale	Balance parameters were significantly improved in the VR training group. This effect was also associated with a significant reduction in falls and lower levels of fear of falling. Some components of balance that were improved by VR training showed a decline after 9 months post-training.
Bacha et al., 2019	[38]	Randomized controlled trial: Kinect Adventures Training Group (KATG) or the conventional physical therapy group (CPTG)	23	23	MBA-p	A	Non-immersive VR	Kinect v2 motion sensor (Microsoft Corporation)	Games (“Space Pop”, “20,000 Leaks”, “Reflex Ridge”, and “River Rush”) that stimulate faster multidirectional movements and center of gravity control, through multidirectional steps, squats, jumps, coordinated movements of the upper and lower limbs, and trunk movements in three planes	Balance: Mini-Balance Evaluation Systems Test (Mini-BESTest); gait: functional gait assessment (FGA); cardiorespiratory fitness: six-minute step test (6MST).	Cognitive functions: Montreal Cognitive Assessment (MoCA)	The results of the study showed that there was no difference between groups regarding the effectiveness to improve balance, gait, cardiorespiratory fitness, and cognition (MoCA) in which both interventions promoted equally positive effects on these outcomes; there was no superiority between the groups.
Campo-Prieto et al., 2022	[39]	Randomized controlled trial: VR intervention group vs. conventional exercising control group	13	11	MBA-p	A	Immersive VR	HTC Vive	An immersive scenario BOX VR (from VIve port library) where they have to simulate boxing techniques (guard, jab, cross, hook, uppercut)	Balance and gait: Tinetti test and the timed up and go test (TUG)	Motion sickness: Simulator Sickness Questionnaire (SSQ) and the System Usability Scale (SUS)	The VR group showed a significant improvement in the Tinetti scores, particularly in the balance section; in addition, there were significant differences between both groups in favor of the intervention group for the TUG test.
Campo-Prieto et al., 2022	[40]	Feasibility trial: VR Intervention group vs. traditional exercise control group	13	11	MBA-p	A	Immersive VR	HTC Vive	BOX VR; three sessions per week, 10 weeks	Physical function: balance and gait: Tinetti test; mobility, lower limb function: timed up and go (TUG) test and the five times sit-to-stand test (FTSTS); hand grip strength (HGS)	Quality of life (QoL): 12-Item Short Form Survey (SF-12); motion sickness: Simulator Sickness Questionnaire (SSQ) and the System Usability Scale (SUS)	The IG showed significant improvements in the Tinetti scores for balance gait, total score, and hand grip (pre- to post-assessment). The CG was significantly worsened compared with the IG in the five times sit-to-stand test, Tinetti scores for balance, gait, and total score, and the Timed up and go test total score (post-assessment).
Yoo et al., 2013	[41]	Randomized controlled trial: augmented reality-based Otago exercise group vs. conventional Otago control group	10	11	MBA-p	A	AR	Computer with a web camera, SVGA resolution head-mounted display (i-visor FX601)	Augmented reality “Otago” exercise: muscle strengthening, balance training, and walking	Balance: Borg balance score, gait function: GAITRite system (GAITRite, CIR systems Inc., Havertown, PA, USA)	Mini Mental State Examination (MMSE); fear of falling (FES-I)	For balance and gait, both groups improved with no differences between groups, while falls efficacy improved only in the intervention group.
Hee Cho et al., 2014	[42]	Randomized controlled trial: VR intervention group vs. pure control group	17	15	MBA-p	B	Non-immersive VR	Wii Balance Board (Nintendo)	Wii Fit virtual reality training, consisting of a balance board and a CD, where when a subject mounts the balance board, an avatar appears on the screen and imitates the subject’s movements; three times a week for eight weeks	Balance: Romberg test, center of pressure excursion (COP) on Bio-rescue (RM INGENERIE)	Mini Mental State Examination (MMSE)	Intervention group showed an improvement on all the physical indexes evaluated, while CG did not.
Ku et al., 2019	[43]	Randomized controlled trial: AR intervention group vs. conventional exercise control group	18	16	MBA-p	A	AR	TETRAX (tetra atriaxametric posturography)	3D interactive Kinect balance exercises: Balloon game for hip exercise, Cave game for knee exercise, and Rhythm game; thrice per week for 1 month	Balance: Berg balance scale (BBS); walking ability: functional ambulation categories (FAC); modified Barthel index (MBI); Fugl Myer-lower extremity score FM L/Ex; Fugl-Meyer motor assessment coordination section (FMA-C), Fugl-Meyer motor assessment balance section (FMA-B), TUG, timed-up-and-go; stability index (SI); weight distribution index (WDI); and automatic balance score using Tetrax posturography	Mini Mental State Examination (MMSE)	A significant group × time (before and after exercise) interaction effect was observed for Berg balance scale (BBS) scores and timed up and go. Overall improvements occurred in the stability index, weight distribution index, fall risk index, and Fourier transformations index of posturography for both groups. However, score changes were significantly greater in the AR intervention group. A significant group × time interaction effect was observed for the fall risk index in favor of the intervention group.
Lee et al., 2017	[44]	Randomized controlled trial: AR group, yoga group, and self-exercise group	10	10 + 10	MPA-np	A	AR	AR-based Otago exercise	AR-based Otago exercise including strengthening balance training, three times a week, 12 weeks	A digital manual muscle tester to measure muscle strength of knee flexion, ankle dorsiflexion, and ankle plantar flexion; strength: manual muscle test (MMT), force plate (FP); fall: Morse fall scale (MFS)	na	In this study, both knee flexion and ankle dorsiflexion strength were significantly increased in all three groups (AR, yoga, and self-Otago exercise groups).Regarding balance, few parameters significantly decreased in the AR group. Regarding falls, the MFS showed significant a decrease in the AR group.
Rebsamen et al., 2019	[45]	Feasibility trial	12	na	MPA-np	C	Non-immersive VR	SENSO (pressure-sensitiveplatform)	An exergame- based activity performed while standing on a pressure-sensitive platform (SENSO), designed as follows: short intervals (one up to 2 min) of higher-intensity exertion “Rocket-game”, alternated with active rest periods for a total of up to 25 min. three times per week, 4 weeks	Adherence, acceptability: Technology Acceptance Model Questionnaire (TAM); usability: System Usability Scale (SUS); enjoyment of exergaming: single five points Likert-like item	Exercise capacity: heart rate at rest (HRrest), heart rate variability (HRV), maximum heart rate (HRmax), and maximum workload (W, in watt) evaluated through maximal exercise testing	Adherence to the HIIT intervention was 91%, and participants showed high acceptance of the intervention considering TAM scores. User satisfaction was rated as excellent, as well as the overall enjoyment of exergaming. The total exercise time mean was 30.8 min. HRmax and HRrest mostly met target ranges. The maximum workload during the incremental exercise test post-training increased significantly over the time.
Fu et al., 2015	[46]	Randomized controlled trial: VR intervention group vs. conventional exercising control group	30	30	MPA-np	A	Non-immersive VR	Wii Balance Board (Nintendo)	Wii Fit balance training group: balance training using Nintendo’s Wii Fit balance board with the games “Soccer Heading”, “Table Tilt”, and “Balance Bubble”; six weeks training	Short-form physiological profile assessment (PPA): visual contrast sensitivity by Melbourne Edge Test; proprioception: lower limb-matching task; quadriceps strength; simple reaction time in milliseconds (using light stimulus), postural sway; fall incidence/risk observed by a nurse		PPA scores and fall incidence improved significantly in both groups after the intervention, but the subjects in the VR training group showed significantly greater improvement in both PPA and falls risk compared with the controls.
de Vries et al., 2020	[47]	Single group	30	na	MPA-np	C	Non-immersive VR	Kinect v2 motion sensor (Microsoft Corporation), Wii Balance Board (Nintendo)	Seven different VR balance games: Wii Sport “Ski”, “Yoga Warrior”, “Yoga”, “Kinect Adventure”, “Kinect Sport”, “Kinect Fitness Yoga”, “Boxing”	Muscle activity of the vastus lateralis, vastus medialis, soleus, and gluteus medius obtained using surface electromyography (EMG)	na	Muscle activity during game play was mostly <40%, and prolonged activation was lacking. Only the games that included more dynamic movements showed higher-intensity muscle activation.
Da Silva Júnior et al., 2021	[48]	Cross-over: participants divided into two groups, the “alone group” vs. the “with peers group”	19	na	MPA-p	C	Non-immersive VR	Kinect v2 motion sensor (Microsoft Corporation)	A bowling exergame played twice a week for 21 weeks	Functional capacity: senior fitness test	Engagement during playtime: Game Experience Questionnaire (GEQ); adherence to PA: Physical Exercise Adherence Questionnaire	After the VR intervention, both groups (alone vs. peers) had significant gains in functional capacity. Comparing the post-test between groups, it was found that the group in which participants played with peers had better outcomes than the group in which participants played alone.
Muñoz et al., 2021	[49]	Observational	57	na	MPA-p	C	Non-immersive VR	Kinect v2 motion sensor (Microsoft Corporation)	Shoulder abduction and the double leg squat; six sessions following VR instructor; over a 15-day period	Physical achievement measured by the Kinect sensors	Perceived usability: System Usability Scale (SUS)	Statistical analysis of survey scores showed a progressive acceptance of the tool by older users, as well as a progressive improvement of physical achievement. Correlations between usability and physical achievements were also found.
Bickmore et al., 2013	[50]	Two-arm, single-blind, randomized controlled trial conducted to compare the intervention group with an active control group (pedometer provided); assessment at 0, 2, and 12 months	132	131	MPA-p	B	Non-immersive VR	Take-home touch-screen tablet	Animated computer fitness instructor that simulates face-to-face conversation using voice, hand gestures, gaze cues, and other nonverbal behavior	Fitness and mobility: timed maximal walking velocity;number of steps: pedometer	Depressive symptoms: PHQ-9; cognitive status: Mini-Cog; health literacy: Test of Functional Health Literacy in Adults (TOFHLA); quality of life: Short Form Survey (SF-12);	Intervention was more effective at increasing fitness and mobility levels than the control group, but no differences emerged at 10 and 12 months after intervention. Health literacy emerged as a moderator of the efficacy of the treatment.
Jeon and Kim, 2020	[51]	Randomized controlled trial: intervention AR group vs. control	13	14	MPA-p	B	AR	UINCARE-HEALTH system	AR interactive program that provides feedback on the correct movement: 30 min program, which includes regular, aerobic, and flexibility exercises; five times a week for 12 weeks	Muscle parameters: skeletal muscle index (SMI) and skeletal muscle mass (ASM); physical performance: gait speed, senior fitness test (SFT); strength: hand grip strength test, chair stand for 30 s, 2 min standing test (2MST); balance: sit and reach test (TUG); walking skill: figure-of-eight walk test (F8W)	Exercise self-efficacy (ESE)	Muscle parameters (SMI and ASM) increased more in the intervention group compared with the control group, and there was a significant increase in gait speed. For physical performance, a significant change was observed in the chair stand test and the timed up-and-go test (TUG), and a significant increase was also observed for exercise self-efficacy.
Barsasella et al., 2021	[52]	Randomized controlled trial: intervention vs. pure control group	29	31	MPA-p	B	Immersive VR	HTC Vive	Nine VR apps: 1. “The Lab”; 2. ”Everest VR”; 3. “The Body VR: Journey Inside a Cell”; 4. “To The Top”; 5. “Waltz of the Wizard”; 6. “Google Earth VR”; 7. “Found”; 8. “Sparc”; 9. “Final Soccer VR”; 15 min twice a week for 6 weeks	Functional fitness: 30 s chair stand test, arm curls, 2 min step test, chair sit and reach test, single-leg standing test, back scratch, 8-foot up and go	Quality of life: (EQ-5D-3L) happiness: mini version of the Chinese Happiness Inventory (CHI)	Quality of life was improved by some metrics assessed (pain/discomfort and anxiety/depression) in both groups. Happiness significantly improved in the intervention group compared with the control group. Among the functional fitness tests, the back scratch test first and back scratch test second significantly improved in the intervention group in comparison with the control group.
Tammy Lin and Wu, 2021	[53]	Cross-over: 2 (avatar age: young vs. older) × 2 (sex: male vs. female) design	104	na	MPA-p	C	Immersive VR	HTC Vive Kinect version 2 motion sensor (Microsoft Corporation), MakeHuman	The participants entered a virtual gym and saw themselves facing a large mirror wall. Then a voice instructed them to perform a series of simple. After, participants were requested to perform seven simple exercises, such as marching in place, tap out, heel down in the front, and chest stretch.	Physical activity: step counts, Borg rating of perceived exertion scale	Self-concept: four items developed for the study; implicit association test (IAT); mood: affect grid;exercise efficacy: self-efficacy for exercise (ESE)	The results showed that the embodiment of younger avatars (age approximately 20 years) in VR leads to greater perceived exercise exertion regardless of sex after controlling for age and emotion. Older adults with young avatars perceived a greater contribution of efforts to exercise. Among those who did not engage in vigorous exercise, female older adults who embodied young avatars reported greater self-efficacy for future exercise and greater physical activity during the exercise phase than those who embodied older avatars.
Lee et al., 2015	[54]	Randomized controlled trial	26	28	MPA-p	A	Non-immersive VR	Kinect v2 motion sensor (Microsoft Corporation)	Three-dimensional avatar with a virtual instructor in a wilderness setting (a lake surrounded by mountains), where participants are asked to follow instructions of the virtual trainer; 60 min intervention three times a week for 8 weeks	Functional ability: 30-s chair stand test (30SCST), 8-foot up-and-go test (8FUGT), and 2- minute step test (2MST)	Quality of life (SF-36); Mini Mental State Examination (MMSE)	IG showed greater improvement in mental health and lower body strength, compared with GG, with mixed results when considering within-group analysis for HRQoL subscales. Both groups showed an increase in all the physical index measures.
Nonino et al., 2018	[55]	Randomized controlled trial: VR intervention group vs. pure control group	12	12	MPA-p	B	Non-immersive VR	Wii console (Nintendo)	Bowling, Wii Sports, 8 weeks	Reported physical activity: International Physical Activity Questionnaire (IPAQ);balance: timed up and go (TUG), baropodometric test, and trail test	Cognitive functions: Mini Mental State Examination (MMSE); satisfaction with life scale	There was significant reduction in the trail test and TUG after the intervention period only for the intervention group; there was a partial significant improvement in few items of the satisfaction with life questionnaire in favor of the intervention group.
Kamińska et al., 2018	[56]	Observational: single group	23	na	MPA-p	C	Non-immersive VR	Kinect v2 motion sensor (Microsoft Corporation)	“Football”, “Bowling”, and “Downhill Skiing” games from Xbox 360 Sports games	Functional ability: the 6 min walking test (6MWT); the Dynamic Gait Index (DGI); the tandem stance test (TST); the tandem walk test (TWT); strength: “spring hand dynamometer”	Depression: Beck Depression Inventory (BDI)	Physical and psychological outcomes were significantly improved. Both groups (under 80 years of age and those aged 80 years and over) had significantly better results on the 6MWT, the TST, and the BDI.
García-Bravo et al., 2020	[57]	Prospective longitudinal study	14	na	MPA-p	C	Non-immersive VR	Wii console (Nintendo), Wii Balance Board (Nintendo)	Wii Fit Plus Software; after the different warm-up exercises (yoga and strengthening exercises), various exercises of reeducation of posture and balance; two sessions per week, 4 weeks	Automatic posturography evaluated different indexes: center of gravity (COG); reaction times (RTs); movement speed (MVL); end point excursion (EPE); maximum excursion (MXE); directional control (DCL)	Quality of life (SF-36)	The results of this study show improvements in the scores of all the posture/physical variables analyzed (i.e., RT, MVL, EPE, MXE, and DCL). Regarding the SF-36 Questionnaire, the results showed statistically significant improvement in vitality and emotional.
Kruse et al., 2021	[58]	Cross-over: traditional, recorded 2D gymnastics video with an immersive virtual reality (VR) exergame	25	na	MPS	C	Immersive VR	Valve Index VR headset	VR intervention based on “Maestro Game VR”, where the players are located in a 3D concert hall and are playing the role of a conductor that has to conduct a band of three musicians by following a virtual 3D path in front of them with a baton	na	Motion sickness: Simulator Sickness Questionnaire (SSQ); motivation: Intrinsic Motivation Inventory (IMI); cognitive functions: MEC spatial presence questionnaire (MEC-SPQ); perceived workload: Nasa-TLX	Both programs received similar scores regarding enjoyment, workload, and attention. However, the heart rate and movement values for the video-based exercise were significantly higher than those for the VR exergame.

Note: N = number of participants; PA = physical activity; VR = virtual reality; AR = augmented reality; IG = intervention group; CG = control group; na = not available.

## Data Availability

No new data were created or analyzed in this study. Data sharing is not applicable to this article.

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
