# Peer review of "Exploring the Psychological Nexus of Virtual and Augmented Reality on Physical Activity in Older Adults: A Rapid Review"

_behavsci, 2023, doi:10.3390/bs14010031_

Round 1

Reviewer 1 Report

Comments and Suggestions for Authors

Thank you for the opportunity to review this paper. The topic is timely as technology is close to ubiquitous use of haptic feedback, AI interactive activity programs and an ever-aging population.

I would like to suggest an organization of the table of reviewed papers. The treatments vary from physical activity movement, game play, workout/fitness conditioning programs, balance training. My suggestion is to group the studies under themes, rather than randomly spread. That way the varied treatments are more readily compared by the reader. Really just a sorting of the table.

Discussion- One area that I think would be of value to provide information is the potential or study-reported side effects of VR program. Often nausea and dizziness is reported when playing games. often this dissipates with exposure, but I did not see much related to any reported side effects of the treatments. if this was reported in the studies, it would be worthy of addressing in the discussion.

Note typo- Reference #1 ...... (2021-2030).; Wolrd Health.....

Author Response

R: Thank you for the opportunity to review this paper. The topic is timely as technology is close to ubiquitous use of haptic feedback, AI interactive activity programs and an ever-aging population.

 I would like to suggest an organization of the table of reviewed papers. The treatments vary from physical activity movement, game play, workout/fitness conditioning programs, balance training. My suggestion is to group the studies under themes, rather than randomly spread. That way the varied treatments are more readily compared by the reader. Really just a sorting of the table.

A: Thanks for your comment. We changed the table according to your comment in order to make it more understandable. Specifically, as you suggested, we tried to give the table of included studies a different structure in order to make it more accessible to our readers and to better convey the heterogeneity of this field of research. In particular, we divided the studies into three main categories. Each of these indicates the kind of treatment implemented in each study. Categories included are MPA (“Mostly physical activity”), MBA (“Mostly balance”) and MPS (“Mostly psychological), and they are sorted in this order. Furthermore, the addition of “p” (psychological) or “np” (no psychological) indicates, respectively, the presence or absence of psychological assessment in the study protocol.

R: Discussion- One area that I think would be of value to provide information is the potential or study-reported side effects of VR program. Often nausea and dizziness is reported when playing games. often this dissipates with exposure, but I did not see much related to any reported side effects of the treatments. if this was reported in the studies, it would be worthy of addressing in the discussion.

A: Thanks for your comment: As for your second suggestion, our lack of focus on study-reported side effects of the AR/VR programs, so we conducted an in-depth analysis in this regard. As a result, we added a new paragraph to our conclusions

R: Note typo- Reference #1 ...... (2021-2030).; Wolrd Health.....

A: Thanks for your comment. We have corrected the error.

Reviewer 2 Report

Comments and Suggestions for Authors

After the inclusion of my suggestions, the paper can be published. 

Comments on the Quality of English Language

English for minor mistakes is OK. 

Author Response

The article needs improvement. I have attached several observations that I hope will be useful  in improving the manuscript.

R: In the title at the end, state the examination method (: Rapid review). If you examine VR and AR interventions, that should also be stated in the title.

A: Thanks for your comment. We changed the title according to your suggestions

R: Line 35: Which decade are you referring to? Please specify.

A: Thanks for your comment. We specified the decade

R: Line 49: State the year of that claim.

A: Thanks for your comment. We specified the year of the claim

R: Line 109: If you examine AR (which you claim), you should explore this theoretical concept in the introduction section.

A: Thanks for your comment. We added a paragraph about  the theoretical concept of AR

R: Line 162: N usually refers to the population size, whereas n refers to the sample size. Please adjust that accordingly.

A: Thanks for your comment. We have corrected the error according to your comment

R: Line 196: immersive vs. non-immersive.

A: Thanks for your comment. We have corrected the error according to your comment.

R: Line 200: Most investigated studies were VR (20+68%). The rest was AR (13%); it is 101%.

A: Thanks for your comment. We added the correct percentages.

R: Line 238-243: Please reformulate this paragraph to make it more understandable.

A: Thanks for your comment. We reformulated this paragraph hoping to be more understandable.

R: Line 324: Liu and colleagues state the year of that study.

A: Thanks for your comment. We added the year of the study conducted by Liu and colleagues

R: Line 330: What does VRR stand for?

A: Thanks for your comment. It was a typo. It should have been VR. We have corrected the error

R: Line 331: mediators or moderators.

A: Thanks for your comment. The role of psychological factors was analyzed both as moderators and mediators in different studies. We have changed the sentence slightly to make it clearer

R: Line 338: state the strengths of your study, if there are any.

A: Thanks for your comment. We added a paragraph about the strengths of our study

R: To make it more understandable, I would clearly distinguish between VR and AR and its influence on the dependent variables - physical activity outcomes and psychological outcomes. Or deal only with one independent variable (VR).

A: Thanks for your comment. We added in the results section several statements in order to clarify the distinction between AR and VR.

R: On the one hand, you examine the effectiveness of VR and AR on the enhancement of physical exercise – activity while at the same time exploring the role of selected psychological variables that, in my view, should have one denominator to make it more understandable and coherent. Your main goal is to know how VR and AR increase physical activity participation. The psychological variables are the outcome of the involvement in VR and AR. So, in that respect, I would envelop psychological consequences to the one or two main common effects (e.g., wellbeing, health, quality of life, etc.).

A: Thanks for your comment. We have tried to make the results section clearer and more consistent by dividing the different outcomes into subsections. In addition, in the results section, we added a specification of the studies at the end of each paragraph to clarify the distinction between AR and VR to highlight the relationships between different technology systems with different psychological variables and physical activity measures.

R: On the other hand, you also explore feasibility studies to determine merits and viability that examine some, not all, aspects of the proposed project. Here, you are opening up other them (validity, technical considerations, and program description). To make it more understandable, I focus on one or two dependent variables that do not open other possibilities. More specifically, it explores mainly the enhancement of physical activity on well-being and quality of life that can be integrally followed with the secondary aim of the psychological benefits of VR and AR participation.

A: Thanks for your comment. We reduced the paragraph focusing on physical and psychological outcomes.

Round 2

Reviewer 2 Report

Comments and Suggestions for Authors

The paper can be published in the present form. 

Author Response

Thank you so much for your time and your availability.